# Reconciling Competing Sampling Strategies of Network Embedding

**Yuchen Yan, Baoyu Jing, Lihui Liu, Ruijie Wang, Jinning Li,**
**Tarek Abdelzaher**, **Hanghang Tong**
University of Illinois at Urbana Champaign, IL, USA
{yucheny5, baoyuj2, lihuil2, ruijiew2, jinning4, zaher, htong}@illinois.edu

## Abstract

Network embedding plays a significant role in a variety of applications. To capture the topology of the network, most of the existing network embedding algorithms follow a *sampling* training procedure, which maximizes the similarity (e.g., embedding vectors' dot product) between positively sampled node pairs and minimizes the similarity between negatively sampled node pairs in the embedding space. Typically, close node pairs function as positive samples while distant node pairs are usually considered as negative samples. However, under different or even competing sampling strategies, some methods champion sampling distant node pairs as positive samples to encapsulate longer distance information in link prediction, whereas others advocate adding close nodes into the negative sample set to boost the performance of node recommendation. In this paper, we seek to understand the intrinsic relationships between these competing strategies. To this end, we identify two properties (*discrimination* and *monotonicity*) that given any node pair proximity distribution, node embeddings should embrace. Moreover, we quantify the empirical error of the trained similarity score w.r.t. the sampling strategy, which leads to an important finding that the *discrimination* property and the *monotonicity* property for *all* node pairs can not be satisfied simultaneously in real-world applications. Guided by such analysis, a simple yet novel model (SENSEI) is proposed, which seamlessly fulfills the *discrimination* property and the *partial monotonicity* within the top-$K$ ranking list. Extensive experiments show that SENSEI outperforms the state-of-the-arts in plain network embedding.

## 1 Introduction

In the era of big data, network embedding [36, 14, 21, 7, 42] maps nodes in the network to low-dimensional vectors in the embedding space, which plays an important role in many tasks such as node recommendation [54, 61, 38, 8, 20, 19, 1, 2], networked time series imputation [44, 18, 48, 9], knowledge graph completion [49, 46, 47, 45, 28], and network alignment [52, 53, 60, 62, 16, 27, 59]. To distill the topology information of the network, most existing network embedding methods follow a *sampling* training procedure. Given any central node to be considered, existing network embedding methods build a positively sampled node pair set and a negatively sampled node pair set. Then, they optimize the embeddings of nodes by maximizing/minimizing the similarity between the positively/negatively sampled node pairs in the embedding space. Explicitly or implicitly, these methods are based on an assumption that nodes close to the central node should be included in the positively sampled node pair set, whereas distant nodes are likely to be considered as negative samples. For example, DeepWalk [36] and metapath2vec [7] explicitly construct random walk containing positively sampled nodes and select negative nodes according to the degree distribution of the network. The message-passing mechanism in GraphSAGE [15] and graph auto-encoder (GAE) [22] is built on graph Laplacian regularization [21, 57], which makes connected node pairs to be

37th Conference on Neural Information Processing Systems (NeurIPS 2023).

similar. In this case, the close/distant nodes become implicit positive/negative samples for the central node, which can be regarded as a general form of such sampling training procedure.

Numerous network embedding algorithms have been proposed, which focus on improving either the positive sampling strategy or the negative sampling strategy based on various intuitions. Consequentially, disparate or even competing sampling strategies emerge in various methods. To name a few, rather than solely favor nodes within two hops as positive samples in LINE [41], node2vec [14] allows a longer random walk length. For graph convolution network (GCN) [21], APPNP [23] utilizes personalized pagerank [33] to sample more distant nodes w.r.t the central node for feature aggregation. The aforementioned two algorithms accommodate farther nodes in the positively sampled node pair set. Meanwhile, SPNE [12], KBGAN [4] and RecNS [56] encourage choosing closer nodes as *difficult* negative samples to promote the performance of node recommendation.

Thus, some fundamental questions arise: *what is the theoretic root cause behind such antithetical sampling strategies? what are possible and what are impossible for sampling of network embedding? how can we develop a practical embedding algorithm that simultaneously embraces these competing strategies?*

In this paper, we hammer at bringing the intrinsic relationships behind these competing sampling strategies in light. Concretely, we start from two fundamental tasks of graph learning: the *link prediction* task and the *node recommendation* task. To tackle the above two tasks synchronously, we identify two desirable properties, including the *discrimination* property and the *monotonicity* property, that network embedding vectors should satisfy given a node pair proximity distribution. The *discrimination* property means that the node pair with high proximity should be distinguished from the node pair owning low proximity in the embedding space (*link prediction*). For the *monotonicity* property, the ranking list of nodes recommended to the query node needs to be consistent with the proximity list in the descending order. Theoretically, we analyze the general form of network embedding algorithms' loss functions. We show that the negative sampling distribution should be negatively correlated with the node pair proximity distribution. Furthermore, we show that in the ideal case where the algorithm can sample an infinite number[1] of positive nodes for the central node, both the *discrimination* property and the *monotonicity* property can be fulfilled (i.e., possibility result). However, due to the limited sample size of the real-world data, there exists an inevitable error between the ideal optimal similarity scores and the empirical optimal similarity scores in the embedding space. Regardless of the specific sampling strategy, the *discrimination* property and the *monotonicity* property for *all* node pairs in the network can not be fulfilled at the same time (i.e., impossibility result).

Fortunately, in real-world applications, ranking all nodes in the network for the query node is often unnecessary. In many cases, only the top-$K$ recommendation list and its internal order matters, which suggests that we can first achieve the *discrimination* property to detect candidate nodes to be recommended. After that, we can attain the *monotonicity* property within the top-$K$ ranking list (*partial monotonicity*). Guided by this intuition and the theoretical results, we propose a simple yet novel model (SENSEI). In detail, SENSEI adopts a commonly used proximity measurement (personalized pagerank [33]) and decomposes the embedding process into two steps. The first step is to satisfy the *discrimination* property. In addition to sampling nodes with large proximity as positive samples, SENSEI also includes nodes with intermediate proximity in the positively sampled node pair set, which reduces the empirical error of the similarity scores for these nodes. Then, in the second step, SENSEI pays attention to the *monotonicity* property within the positively sampled node pairs from the first step. Nodes are ranked w.r.t. their proximities and importantly, some positively sampled nodes are turned into negative samples, which resembles the strategy of selecting *difficult* negative samples in some existing methods [4, 55]. In this way, SENSEI creatively integrates these two competing sampling strategies in one integral framework. The experiments demonstrate that SENSEI greatly outperforms various baselines.

To summarize, our contributions are three folds:

- **Theoretical Analysis.** We reveal the underlying relationships between the competing sampling strategies of existing network embedding methods. Specially, we prove that any

---

[1]Instead of having an infinite number of nodes in the network, we could sample positive pairs infinite times to approximate the true distributions.

sampling strategy bears an inevitable error gap between the empirical and ideal optimal embedding similarity scores.

- **Simple yet Novel Model.** Based on the theoretical results, we propose a two-step model SENSEI which creatively integrates competing sampling strategies in one network embedding framework. It satisfies the *discrimination* property in the first step and obtains the *partial monotonicity* property within the positively sampled node pair set in the second step.
- **Experimental Results.** Extensive experiments show that SENSEI outperforms the state-of-the-arts in plain network embedding.

**Problem setting.** We primarily focus on plain network embedding, in which given the adjacency matrix of a plain network $\mathbf{A}$, we aim to output the embedding matrix $\mathbf{F}$.

## 2 Analysis

In this section, we uncover the intrinsic relationships of existing competing sampling strategies. First, we describe the sampling-based network embedding process mathematically. Then, we propose two desired properties for the learned embeddings: the *discrimination* property and the *monotonicity* property, which correspond to two common learning tasks: *link prediction* and *node recommendation*. Theoretically, we analyze the general form of network embedding algorithms' loss functions. We show that the negative sampling distribution should be *negatively* correlated with the given node pair proximity distribution. Guided by this critical insight, we give both possibility and impossibility results for network embedding. First (possibility result), we show that, in the ideal case where an infinite number of positive samples can be obtained, both the *discrimination* property and the *monotonicity* property can be satisfied with the optimal solution of the loss function. Second (impossibility result), for the empirical loss function with a limited number of positive samples, we prove that there always exists an error gap between the empirical optimal solution and the ideal optimal solution. Regardless of the specific sampling strategy, the *discrimination* property and the *monotonicity* property can not be simultaneously satisfied for *all* node pairs in the network in the empirical situation.

Current sampling-based network embedding algorithms can be divided into two phases. The first phase is to construct a node pair proximity distribution $p(u|v)$, which serves as the positive sampling distribution in the training process. For example, personalized pagerank [33] in APPNP [23] is an explicit node pair proximity distribution, while random walk in DeepWalk [36] and node2vec [14] implicitly constructs such node pair proximity distribution. The second phase is to design a negative sampling distribution $p_n(u|v)$ and to utilize the network embedding loss function to train the embedding model. Similar to previous works [55, 56], we start from analyzing the second phase of the sampling-based network embedding process as follows:

**Definition 1.** *Sampling-based Network Embedding Process in the Second Phase. Given a node proximity distribution $p$, where $p(u|v)$ refers to the proximity of node $u$ w.r.t. the central node $v$ and $\sum_u p(u|v) = 1$, the algorithm designs a negative sampling distribution $p_n$ and adopts a loss function $J$ to obtain similarity scores $s(u,v) \in [0,1]$ of all node pairs $(u,v)$, which are calculated by the node embedding matrix $\mathbf{F}$.*

The similarity score $s(u,v)$ defined here is general, which can be implemented with various similarity measurements (e.g., the sigmoid function $\sigma(\mathbf{F}(u,:)^\top \mathbf{F}(v,:))$ or the cosine function $cos(\mathbf{F}(u,:), \mathbf{F}(v,:))$) in various algorithms.

Then, to solve both link prediction and node recommendation tasks, we propose that the general similarity score $s(u,v)$ calculated by node embeddings should possess two properties: the *discrimination* property and the *monotonicity* property, which are defined as the following:

**Definition 2.** *Discrimination and Monotonicity.*

*Discrimination Property: Given the central node $v$, node pair $(u,v)$ with large $p(u|v)$ should be clearly distinguished with node pair $(w,v)$ with small $p(w|v)$ in the embedding space, i.e., $\lim \frac{s(u,v)}{s(w,v)} = +\infty$ when $\frac{p(w|v)}{p(u|v)} \to 0$.*

*Monotonicity Property: Given the central node $v$ and two arbitrary nodes $u$, $w$ in the network, if $p(u|v) > p(w|v)$, then $s(u,v) > s(w,v)$.*

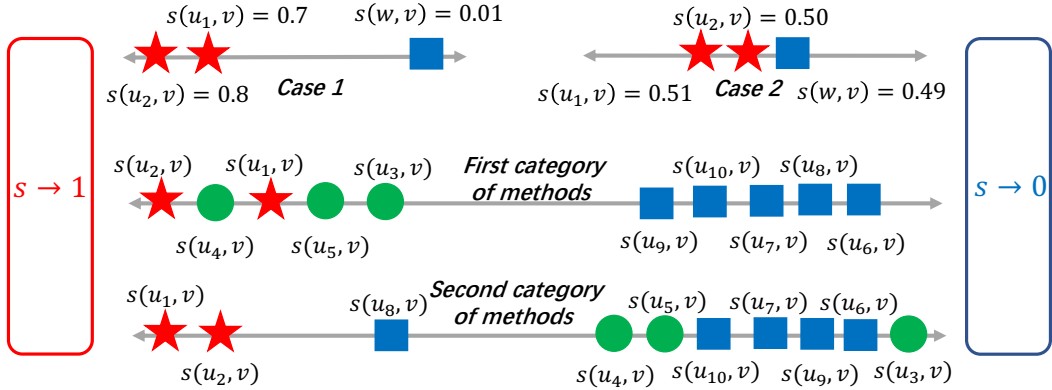

Figure 1: ***Discrimination*** **and** ***monotonicity*** **properties**. We use star nodes to denote nodes with large $p(u|v)$, round nodes to denote nodes with intermediate $p(u|v)$ and rectangle nodes to denote nodes with small $p(u|v)$. We assume that $p(u_1|v) > p(u_2|v) > \cdots > p(u_{10}|v)$.

The *discrimination* property bears subtle difference from the *monotonicity* property. Let us look at an illustrative example in the first line in Figure 1. For the central node $v$, we assume that $p(u_1|v) = 0.45$, $p(u_2|v) = 0.4$ and $p(w|v) = 0.001$. ***Case 1***: If $s(u_1, v) = 0.7$, $s(u_2, v) = 0.8$ and $s(w, v) = 0.01$, the final node embeddings satisfy the *discrimination* property ($\frac{s(w,v)}{s(u_1,v)} = 0.0143$ and $\frac{s(w,v)}{s(u_2,v)} = 0.0125$, both approaching zero) but do not satisfy the *monotonicity* property ($p(u_1|v) > p(u_2|v)$ but $s(u_1, v) < s(u_2, v)$). ***Case 2***: If $s(u_1, v) = 0.51$, $s(u_2, v) = 0.50$ and $s(w, v) = 0.49$, the final node embeddings fulfill the *monotonicity* property ($s(u_1, v) > s(u_2, v) > s(w, v)$) but do not satisfy the *discrimination* property ($\frac{s(u_1,v)}{s(w,v)} = 1.04$ and $\frac{s(u_2,v)}{s(w,v)} = 1.02$). From the above example, we can see that the *discrimination* property focuses on making node pairs with large proximity closer and pushing away node pairs with small proximity in the embedding space, which is desirable for tasks like binary classification for link prediction (i.e., predicting the existence of a link between two nodes). On the other hand, the *monotonicity* property pays more attention to the rank of the similarity scores of node pairs, which is critical for tasks like ranking for node recommendation.

## 2.1 Possibility Results in the Ideal Case

We conduct theoretic analysis about the loss function of network embedding. Previous works [24, 37, 55] have demonstrated that most existing network embedding algorithms can be regarded as implicit matrix factorization and there exists a general form of loss function for the given central node $v$. If the sigmoid function is adopted as the similarity function. The loss function is:

$$J = -\mathbb{E}_{u \sim p(u|v)} \log \sigma(\mathbf{F}(u,:)^\top \mathbf{F}(v,:)) - k\mathbb{E}_{w \sim p_n(w|v)} \log(1 - \sigma(\mathbf{F}(w,:)^\top \mathbf{F}(v,:))) \quad (1)$$

where $\sigma(\cdot)$ is the sigmoid function, $\mathbf{F}(u,:)$, $\mathbf{F}(v,:)$ and $\mathbf{F}(w,:)$ are the embeddings of $u, v$ and $w$ respectively, and $k$ is the number of negative samples for each positive sample. If the sigmoid function is replaced by other similarity measurements, we can obtain a more general form of this loss function:

$$J = -\mathbb{E}_{u \sim p(u|v)} \log(s(u, v)) - k\mathbb{E}_{w \sim p_n(w|v)} \log(1 - s(w, v)) \quad (2)$$

where $s(\cdot, \cdot)$ is a general form of similarity measurement.

In the ideal case where an infinite number of positive samples from $p(u|v)$ are available, we have the following theorem about the optimal solution of $s$:

**Theorem 1.** ***Optimal Solution of*** $s$. *The optimal solution of the similarity function $s$ satisfies that for each node pair $(u, v)$,*

$$s(u, v) = \frac{p(u|v)}{p(u|v) + kp_n(u|v)} \quad (3)$$

*Proof.*

$$J = -\mathbb{E}_{u \sim p(u|v)} \log s(u,v) - k\mathbb{E}_{w \sim p_n(w|v)} \log(1 - s(w,v)) \tag{4}$$

Since $s(u,v)$ is the parameter to be optimized, we can calculate $\nabla_{s(u,v)} J$ as following:

$$\nabla_{s(u,v)} J = -p(u|v)\frac{1}{s(u,v)} - kp_n(u|v)\frac{1}{s(u,v) - 1} \tag{5}$$

Let the derivative to be 0, we can obtain $s(u,v) = \frac{p(u|v)}{p(u|v)+kp_n(u|v)}$, which is the optimal solution for $J$. $\qquad\square$

With the optimal solution of $s(u,v)$, we give the following proposition about the design of $p_n$:

**Proposition 1.** *The Design of Negative Sampling. In the sampling-based network embedding process, the negative sampling distribution $p_n$ should be **negatively** correlated with the given node pair proximity distribution $p$ to fulfill the monotonicity property.*

*Proof.* Here we prove the above proposition in the ideal case by contradiction: In the ideal case, assume that the best embeddings and corresponding similarity scores to satisfy the *discrimination* property and the *monotonicity* property are obtained with $p_n$, and $p_n$ is not negatively correlated with $p$. It is equivalent to the situation that for one central node $v$, there must exist two nodes $u, w$ with $p(u|v) > p(w|v)$ and $p_n(u|v) > p_n(w|v)$. From Theorem 1 we know that $s(u,v) = \frac{p(u|v)}{p(u|v)+kp_n(u|v)}$ and $s(w,v) = \frac{p(w|v)}{p(w|v)+kp_n(w|v)}$. Now, we interchange $p_n(u|v)$ with $p_n(w|v)$ and we can get new similarity scores $s'(u,v)$ and $s'(w,v)$ for $(u,v)$ and $(w,v)$. $s'(u,v) = \frac{p(u|v)}{p(u|v)+kp_n(w|v)} > \frac{p(u|v)}{p(u|v)+kp_n(u|v)} = s(u,v)$. In addition, $s'(w,v) = \frac{p(w|v)}{p(w|v)+kp_n(u|v)} < \frac{p(w|v)}{p(w|v)+kp_n(w|v)} = s(w,v)$. So, we get two new similarity scores that can satisfy the *monotonicity* property better, which is contradictory with the assumption that we have already achieved the best embeddings and corresponding similarity scores. $\qquad\square$

Intuitively, this proposition makes sense because if one node is more likely to be sampled as a positive sample for the central node, it is less likely that this node also acts as a negative sample.

From Theorem 1, we can see that if $p_n$ is negatively correlated with $p$, when $p(u|v)$ is large and $p_n(u|v)$ is small, $s(u,v) \to 1$; and when $p(u|v)$ is small and $p_n(u|v)$ is large, $s(u,v) \to 0$, which satisfies the *discrimination* property. For all nodes in the network with $p(u_1|v) > p(u_2|v) > \cdots > p(u_n|v)$, if we set $p_n(u_1|v) < p_n(u_2|v) < \cdots < p_n(u_n|v)$, the *monotonicity* property can be fulfilled with $s(u_1,v) > s(u_2,v) > \cdots > s(u_n,v)$.

## 2.2 Impossibility Results in the Empirical Case

However, in real-world applications, we can *not* sample an infinite number of nodes from $p(u|v)$. Empirically, existing algorithms set a fixed number $T$ of positive samples for the central node and the general loss function in Eq. (2) turns into:

$$J_e = -\frac{1}{T}\sum_{i=1}^{T} \log(s(u_i,v)) - \frac{1}{T}\sum_{i=1}^{kT} \log(1 - s(w_i,v)) \tag{6}$$

where $u_i$ is a positive sample from $p(u|v)$ and $w_i$ is a negative sample from $p_n(w|v)$. We use $S = [s(u_1,v), s(u_2,v), \ldots, s(u_n,v)]$ to denote the final similarity scores to be optimized w.r.t. the central node $v$. For the general loss function in the ideal case, the optimal solution of $S$ is $S^*$, where $S^* = [\frac{p(u_1|v)}{p(u_1|v)+kp_n(u_1|v)}, \ldots, \frac{p(u_n|v)}{p(u_n|v)+kp_n(u_n|v)}]$. For the loss function $J_e$ in the empirical case with $T$ positive samples from $p$, we represent the optimal solution as $S_e$. For the mean squared error between $S^*$ and $S_e$, we have the following theorem:

**Theorem 2.** *Mean Squared Error. For the mean squared error between the empirical optimal solution $S_e$ and the ideal solution $S^*$, we have*

$$\mathbb{E}[||(S_e - S^*)_u||^2] = \frac{1}{T(2+\frac{kp_n(u|v)}{p(u|v)}+\frac{p(u|v)}{kp_n(u|v)})^2}\left(\frac{1}{p(u|v)} + \frac{1}{kp_n(u|v)} - 1 - \frac{1}{k}\right), \tag{7}$$

*where $\mathbb{E}$ is the expectation and $(S_e - S^*)_u = s_e(u,v) - s^*(u,v)$ is the error between the optimal similarity scores obtained in the empirical and ideal cases respectively.*

The proof for Theorem 2 is attached in Appendix A.1. If the sigmoid function is adopted as the similarity function, the expectation of the similarity error in Theorem 2 degenerates to its special case:

$$\mathbb{E}[||\mathbf{F}_e(u,:)^\top \mathbf{F}_e(v,:) - \mathbf{F}^*(u,:)^\top \mathbf{F}^*(v,:)||^2] = \frac{1}{T}(\frac{1}{p(u|v)} + \frac{1}{kp_n(u|v)} - 1 - \frac{1}{k}), \qquad (8)$$

where $\mathbf{F}_e(u,:)$ and $\mathbf{F}^*(u,:)$ are the optimal embedding vectors of node $u$ in $J_e$ and $J$. Eq. (8) was first discovered in MCNS [55]. Based on Eq. (8), MCNS advocates that, in order to bound the dot product error for nodes with large $p(u|v)$, $p_n$ should be *positively* correlated with $p$. Unfortunately, this is not always the best choice due to the following reason. In MCNS, the authors set $p_n(u|v) \propto p(u|v)^a$, where $0 < a < 1$. With this negative sampling distribution, the right part of Eq. (8) changes into $\frac{1}{T}(\frac{1}{p(u|v)}(1 + \frac{p(u|v)^{1-a}}{c}) - 1 - \frac{1}{k})$, where $c$ is a constant. If $p(u|v)$ is large, this term can indeed be bounded. However, for distant nodes with small $p(u|v)$ (i.e., $p(u|v) \to 0$), its positively correlated negative sampling probability $p_n(u|v)$ is also very small and $p_n(u|v) \to 0$, which leads to an extremely large error considering that the term $\frac{1}{p(u|v)} + \frac{1}{kp_n(u|v)} \to \infty$, which may fail to fulfill the *discrimination* property.

***Discussion***. Let us analyze the implications of Theorem 2. If $p_n(u|v)$ is negatively correlated with $p(u|v)$, it means that $\frac{kp_n(u|v)}{p(u|v)} \to 0$ for nodes with the largest $p(u|v)$s and $\frac{kp_n(u|v)}{p(u|v)} \to \infty$ for nodes with the smallest $p(u|v)$s. Let us consider the case that $p(u|v)$ is large and $\frac{kp_n(u|v)}{p(u|v)} \to 0$.
$\mathbb{E}[||(S_e - S^*)_u||^2] < \frac{1}{T(\frac{p(u|v)}{kp_n(u|v)})^2}(\frac{1}{p(u|v)} + \frac{1}{kp_n(u|v)} - 1 - \frac{1}{k}) < \frac{1}{T}(\frac{kp_n(u|v)}{p(u|v)})^2(\frac{1}{p(u|v)} + \frac{1}{kp_n(u|v)})$.
It can be rewritten as $\frac{1}{T}((\frac{kp_n(u|v)}{p(u|v)})^2 \frac{1}{p(u|v)} + \frac{kp_n(u|v)}{p(u|v)} \frac{1}{p(u|v)})$. Since $p(u|v)$ is among the largest proximity scores, $\frac{1}{p(u|v)}$ is approximately bounded by a finite number $\frac{1}{n}$, where $n$ is the number of nodes in the network. As a result, $\frac{kp_n(u|v)}{p(u|v)} \frac{1}{p(u|v)} \to 0$ and $(\frac{kp_n(u|v)}{p(u|v)})^2 \frac{1}{p(u|v)} \to 0$. Therefore, for nodes with large $p(u|v)$, the empirical embedding similarity score $s_e(u,v) \approx s^*(u,v)$. With a similar analysis on nodes with small $p(u|v)$, we reach the same conclusion that $s_e(u,v) \approx s^*(u,v)$. Based on the above analysis, we can see that the empirical optimal solution can achieve the *discrimination* property with a negatively correlated $p_n$: for node pairs with large/small $p(u|v)$, their similarity scores approach the ideal optimal solution. However, there might exist some nodes with $p(u|v) \approx kp_n(u|v)$, which are referred to as nodes with *intermediate* $p(u|v)$. For these nodes, the error in Theorem 2 is larger than nodes with small/large $p(u|v)$ because $(2 + \frac{kp_n(u|v)}{p(u|v)} + \frac{p(u|v)}{kp_n(u|v)})^2$ achieves its minimum when $p(u|v) = kp_n(u|v)$. This can also be explained intuitively as follows. Nodes with large $p(u|v)$ and large $p_n(u|v)$ (small $p(u|v)$) are always sampled as positive samples and negative samples, while nodes with intermediate $p(u|v)$ and $p_n(u|v)$ might be ignored and are less likely to be sampled as positive samples or negative samples. As such, the embedding similarity scores for these nodes bear more uncertainty and large error between the empirical optimal solution and the ideal optimal solution (Theorem 2).

With these findings, we can now unveil the underlying reason for the competing sampling strategies in existing methods. The first category of methods (e.g., [36, 14, 23]) champion sampling nodes with longer distance from the given central node as positive samples. The essence of this positive sampling strategy is to include more nodes with intermediate $p(u|v)$ in the positively sampled node pair set. In this way, these methods realize the goal to minimize the error in Theorem 2 for these nodes with intermediate $p(u|v)$. However, this design has cost. When choosing more nodes with intermediate $p(u|v)$ as positive samples, it implicitly decreases the $p(u|v)$ for nodes with large $p(u|v)$ (since $\sum_u p(u|v) = 1$). As shown on the second line in Figure 1, it hurts the *monotonicity* between these nodes (round nodes) and nodes with large $p(u|v)$ (star nodes). On the contrary, for the second category of methods (e.g., [4, 56, 55]), they prefer to select nodes with intermediate $p(u|v)$ as *difficult* negative samples in the recommendation task, they successfully fulfill the *monotonicity* property between nodes with large $p(u|v)$ (star nodes) and nodes with intermediate $p(u|v)$ (round nodes). Unfortunately, as our discussion about Eq. (8) has demonstrated, this *positively* correlated negative sampling distribution could make the empirical error of the embedding similarity scores of nodes with small $p(u|v)$ quite large. It does harm the *discrimination* property, which is shown on the last line in Figure 1 (e.g., $s(u_8, v)$).

To conclude, due to the limited number $T$ of positive samples, the empirical loss tends to sample nodes with large $p(u|v)$ or $p_n(u|v)$ and there always exist some nodes that can not be sampled as positive/negative samples and thus are ignored. Therefore, the *discrimination* property and the *monotonicity* property for *all* node pairs in the network can not be satisfied simultaneously in the empirical situation.

## 3 Model

In many real-world applications, given the query node, we primarily care about the top-$K$ recommendation list. This suggests that we only need to satisfy the *discrimination* property and the *partial monotonicity* property within nodes that are likely to appear in the top-$K$ recommendation list. Guided by this and the theoretical results in Section 2, we propose a simple yet novel model named SENSEI.

**Key Idea.** The key idea of SENSEI is to seamlessly integrate two competing sampling strategies into one model together. Concretely, it means that we can decompose the proposed SENSEI into two steps. The first step is to satisfy the *discrimination* property and the second step is to achieve the *monotonicity* property within nodes that are likely to appear in the top-$K$ recommendation list. As shown in Figure 2, in the first step, SENSEI samples nodes with intermediate $p(u|v)$, which is the strategy by the first category of methods mentioned in Section 2. SENSEI constructs a *combined* positive sample set, including nodes with large $p(u|v)$ and nodes with intermediate $p(u|v)$. During the training process, SENSEI treats these nodes equivalently as positive samples and maximizes their similarity scores with the given central node. At the same time, it minimizes the similarity scores for nodes with small $p(u|v)$ (large $p_n(u|v)$) to fulfill the *discrimination* property. In the second step, SENSEI focuses on the *difficult* negative samples, which is the strategy by the second category of methods in Section 2. SENSEI pays attention to the *monotonicity* property within the *combined* positive sample set in the first step. In detail, the previously sampled positive nodes have two roles: they act as the positive samples compared to nodes with smaller $p(u|v)$ in the first step and meanwhile function as the *difficult* negative samples compared to nodes with larger $p(u|v)$ in the second step.

**Details.** For $p(u|v)$, we run personalized pagerank [33], which is used throughout SENSEI. In addition, we normalize node embeddings to have unit L2 norm and adopt the dot product $(\mathbf{F}(u,:)^\top \mathbf{F}(v,:))$ as the similarity function $s(u,v)$.

**Step 1: Fulfill the *Discrimination* Property.** In this step, to satisfy the *discrimination* property, we construct a *combined* positive sample set $\mathcal{P}(v)$ for $v$ in $\mathbf{A}$, which includes nodes with large and intermediate $p(u|v)$. Specifically, we rank the proximity $p(u|v)$ for all nodes in $\mathbf{A}$ to obtain a descending list $[p(u_1|v), p(u_2|v), \ldots, p(u_n|v)]$. Then, we set a threshold $\tau$. Nodes with $p(u|v) > \tau$ form the *combined* positive sample set $\mathcal{P}(v)$ for the central node $v$ and the remaining nodes are added into the set $\mathcal{N}(v)$, where negative samples will be randomly selected. So, the loss for node $v$ is:

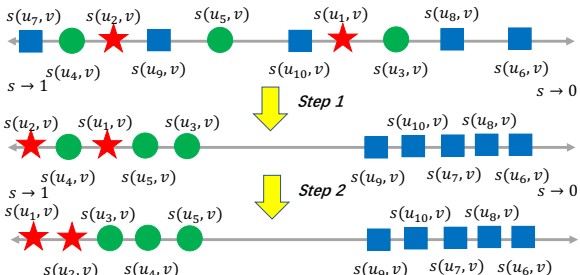

Figure 2: **The training process of SENSEI.** Similar to Figure 1, we still utilize star nodes to denote nodes with large $p(u|v)$, round nodes to denote nodes with intermediate $p(u|v)$ and rectangle nodes to denote nodes with small $p(u|v)$. In Step 1, SENSEI includes round nodes (nodes with intermediate $p(u|v)$) as positive samples. In Step 2, the round nodes are used as hard negative samples for the star nodes.

$$J(v) = -\frac{1}{|\mathcal{P}(v)|}\Big(\sum_{u \in \mathcal{P}(v)} (k\mathbf{F}(u,:)^\top \mathbf{F}(v,:) - \sum_{m=1}^{m=k} \mathbf{F}(w_m,:)^\top \mathbf{F}(v,:))\Big) \quad (9)$$

where $k$ is the negative sample number for each positive sample, $u$ is the positively sampled node for $v$ and $w_m \in \mathcal{N}(v)$ is the negatively sampled node. By adding the loss for all $v$s in $\mathbf{A}$, we obtain the

overall loss in Step 1 as:

$$J_P = \sum_{v \in \mathbf{A}} J(v) \tag{10}$$

**Step 2: Fulfill the *Partial Monotonicity* Property.** In this step, we focus on the *partial monotonicity* property within the *combined* positive sample set in Step 1. For any two nodes $u_l$ and $u_m$ in the positive sample set $\mathcal{P}(v)$, if $p(u_l|v) > p(u_m|v)$, we want to retain the *monotonicity* between them with $\mathbf{F}(u_l,:)^\top \mathbf{F}(v,:) > \mathbf{F}(u_m,:)^\top \mathbf{F}(v,:)$. Node $u_m$ changes from a positive sample for $v$ in Step 1 to a negative sample for $v$ compared with $u_l$, which is consistent with the idea of sampling *difficult* negative samples. The margin-based ranking loss is:

$$L(v) = \sum \mathbb{1}_{p(u_l|v)>p(u_m|v)}(\max\{\mathbf{F}(u_m,:)^\top \mathbf{F}(v,:) - \mathbf{F}(u_l,:)^\top \mathbf{F}(v,:) + \gamma, 0\}) \tag{11}$$

where $u_l, u_m \in \mathcal{P}(v)$, $\mathbb{1}_{p(u_l|v)>p(u_m|v)}$ is the indicator function and $\gamma$ is a positive margin. When $p(u_l|v) > p(u_m|v)$, $\mathbb{1}_{p(u_l|v)>p(u_m|v)} = 1$, otherwise, $\mathbb{1}_{p(u_l|v)>p(u_m|v)} = 0$. Similar to Eq. (10) in Step 1, we have the loss $L_p$ in Step 2 as:

$$L_P = \sum_{v \in \mathbf{A}} L(v). \tag{12}$$

In fact, Step 2 can be regarded as a fine-tuning process to fulfill the *partial monotonicity* property. Therefore, this step usually has a small learning rate and a small positive margin $\gamma$. The pseudocode of this algorithm can be found in Appendix A.2.

**Complexity Analysis.** We give a complexity analysis of SENSEI here. For SENSEI, the time complexity can be analyzed as the following: (1) Calculating personalized pagerank has the complexity $O(\text{iter}_{\max} \cdot \|\mathcal{E}\|)$, where $\|\mathcal{E}\|$ is the number of edges in the graph and $\text{iter}_{\max}$ is the maximum iterations or hops for personalized pagerank; (2) the time complexity for sorting the proximity distribution $p(u|v)$ and sampling positive/negative nodes for node $v$ is $O(n \cdot \log(n))$; (3) Calculating $J(v)$ takes $O(\|\mathcal{P}(v)\| \cdot k \cdot d)$, where $\|\mathcal{P}(v)\|$ is the positive sample set, $k$ is the number of negative samples and $d$ is the dimension of node embedding. Obtaining $J_P$ takes $O(n)$ time; (4) Computing $L(v)$ in Eq. (11) takes $O(\|\mathcal{P}(v)\|^2 \cdot d)$ and calculating $L_p$ costs $O(n)$. Most of these computations can be further parallelized.

## 4 Experiment

In this section, we evaluate the effectiveness of the proposed algorithm (SENSEI) for solving link prediction and node recommendation simultaneously in plain networks.

### 4.1 Experimental Setup

**Datasets.** We use 4 public real-world datasets to evaluate the proposed SENSEI model: C.ele [51], Cora [39], Citeseer [39], NS [31].

**Baselines.** We compare the proposed SENSEI with the following 5 plain network embedding methods: node2vec [14], VGAE [22], GAT [43], ARGVA [35] and RBGE [17].

**Metrics.** For link prediction and node recommendation in plain networks, we randomly split edges in every dataset into 70/10/20% for training, validation, and test.[2] The same amount of additionally sampled non-existent edges are taken as negative edges for training, validation and test in the link prediction task.

**Settings.** For the link prediction task in plain networks, we use Area Under the ROC and Precision-Recall Curves (i.e., AUC-ROC and AUC-PR) as the metrics to evaluate the performance of different methods. For the node recommendation task in plain networks, we take Hit@$K$ as the metric. Only when the model ranks the correct node as top-$K$ in the recommendation list, it is counted as a hit for this query node in the test set. Therefore, the average hit number for all query nodes is Hit@$K$. We set $K = 10$ for SENSEI.

---

[2]The node recommendation task in plain networks is to recommend potential nodes that are most likely to have edges connected to the query node.

**Additional Contents.** In Appendix, we provide more contents related to experiments, including (1) dataset statistics (Appendix A.3); (2) implementation details (Appendix A.3); (3) the comparison with additional GCN-based methods: GraphSAGE [15], Cluster-gcn [5], APPNP [13], GPRGNN [6] and H2GCN [63] (Appendix A.4).

## 4.2 Effectiveness of SENSEI

Table 1: The AUC-ROC ($\pm std$) and AUC-PR ($\pm std$) of link prediction in plain networks (%).

| Models | Cora | | Citeseer | | NS | | C.ele | |
|---|---|---|---|---|---|---|---|---|
| | AUC-ROC | AUC-PR | AUC-ROC | AUC-PR | AUC-ROC | AUC-PR | AUC-ROC | AUC-PR |
| node2vec[14] | 75.96±1.18 | 82.73±0.69 | 69.63±0.97 | 77.38±0.89 | 86.48±0.76 | 91.36±0.75 | **79.36±1.14** | 74.27±1.33 |
| VGAE [22] | 78.28±0.53 | 81.07±0.59 | 72.58±0.74 | 78.00±0.40 | 86.45±0.95 | 89.99±0.58 | 77.78±1.39 | 73.50±2.38 |
| GAT [43] | 76.48±0.77 | 78.90±1.03 | 71.79±1.40 | 76.04±1.18 | 85.89±0.84 | 88.47±1.92 | 74.16±4.33 | 71.21±1.69 |
| ARGVA [35] | 75.99±0.97 | 78.11±1.14 | 71.96±1.10 | 74.74±0.59 | 88.55±1.12 | 90.32±1.36 | 77.04±1.71 | 71.29±2.42 |
| RBGE [17] | 79.33±0.95 | 83.11±0.86 | 73.89±1.11 | 77.48±1.30 | 83.98±4.21 | 88.77±2.77 | 74.65±4.54 | 72.07±3.70 |
| SENSEI | **81.27±0.65** | **83.59±1.11** | **76.28±0.90** | **82.02±0.79** | **91.42±1.25** | **93.10±0.77** | 77.13±0.81 | **77.28±1.53** |

The results of link prediction on all 4 plain networks are presented in Table 1. Our proposed SENSEI generally outperforms all baselines on all 4 datasets. In particular, SENSEI achieves about 2% improvement in AUC-PR compared to the best competitor (e.g., VGAE on Citeseer and node2vec on C.ele). For the node recommendation task, as shown in Table 2, node2vec obtains the best Hit@10 on Cora and NS among all baselines, which indicates that including more positive samples ($T$) by setting a larger walk length (80) can indeed reduce the error in Theorem 2. Notice that the proposed SENSEI still outperforms node2vec on all 4 datasets, which demonstrates the ability of SENSEI to fulfill the *partial monotonicity property*.

Table 2: Hit@10 ($\pm std$) of node recommendation in plain networks (%).

| Models | Cora | Citeseer | NS | C.ele |
|---|---|---|---|---|
| node2vec[14] | 19.78±0.84 | 18.50±1.30 | 59.88±2.68 | 9.43±2.41 |
| VGAE [22] | 8.19±1.62 | 8.93±0.91 | 37.26±2.36 | 9.34±1.32 |
| GAT [43] | 5.77±0.64 | 5.99±2.25 | 23.96±5.39 | 9.71±3.00 |
| ARGVA [35] | 5.65±1.22 | 4.71±0.56 | 36.78±5.20 | 9.71±2.39 |
| RBGE [17] | 17.19±1.24 | 18.99±1.17 | 52.48±5.18 | 9.48±4.86 |
| SENSEI(w/o Step 2) | 19.82±1.18 | 18.17±1.31 | 61.01±4.22 | 11.07±2.18 |
| SENSEI | **20.51±1.13** | **19.03±1.50** | **61.31±3.46** | **11.52±2.13** |

## 4.3 Ablation Study and Sensitivity Study

In this subsection, we conduct the ablation study to validate the importance of Step 2 of SENSEI. As illustrated in Table 2, for the node recommendation task (Hit@10), Step 2 consistently improves the performance of SENSEI. The results verify that Step 2 is indeed beneficial by fine-tuning node embeddings to satisfy the *partial monotonicity* property. In addition, we carry out a sensitivity study on different $K$s for Hit@$K$, which is presented in Figure 3. We observe that when $K$ increases from 1 to 30, Hit@$K$ gradually increases and Hit@30 is much higher than Hit@1, which is expected.

## 5 Related works

**Network Embeddings.** Network embedding maps nodes in the network to low dimensional vectors. It can be traced back to matrix low rank approximation [26] and spectral clustering [40]. DeepWalk [36] and node2vec [14] rely on random walk to encode the topological information of homogeneous networks. For multi-layered networks or heterogeneous networks, metapath2vec [7] and Hin2vec [10] design metapaths to capture the connectivity between different layers or different types of nodes. Since graph convolutional network (GCN) [21] emerges, many GCN-based network embedding methods have been proposed. For example, variational graph auto-encoder (VGAE) [22] adopts the same message-passing mechanism as GCN to embed the homogeneous network. More recently, GCN has been integrated with random walk based heterogeneous network embedding methods, which leads to heterogeneous graph convolutional networks like MAGNN [11] and HAN [50]. MANE [25] successfully employs network embedding on multi-layered networks and DMGC [29] accomplishes both network embedding and clustering by utilizing cross-layer links as regularization. In addition, many embedding algorithms for knowledge graph have been proposed such as translational distance model [3] and semantic matching model [32].

**Sampling in Network Embedding.** Sampling is an important technique, which appears in recommendation [34] and text embedding [30] to speed up the training process. For network embedding algorithms, both the positive sampling strategy and the negative sampling strategy have been applied and improved. For matrix factorization [26], spectral clustering [40] and LINE [41], direct or two-hop neighboring nodes act as positive samples and negative nodes are sampled uniformly or according to the degree distribution. In DeepWalk [36], node2vec [14] and metapath2vec [7], the positive sampling strategy is modified based on truncated random walk starting from the central node. The positive and negative sampling are implemented implicitly with a message-passing mechanism in GCN-based methods, where nodes to be aggregated can be viewed as positive samples and the remaining nodes are implicit negative samples. For the GCN related works, efforts are mainly made

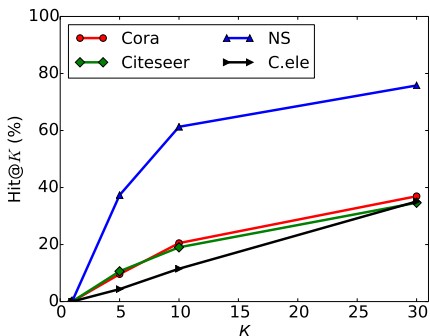

Figure 3: Hit@$K$ for SENSEI.

to improve the positive sampling strategy. For example, APPNP [13] manipulates positive samples with personalized pagerank [33] to catch long distance information and PGNN [58] randomly fixes an anchor node set for aggregation in each training epoch. Recently, more attention has been paid to improve the negative sampling strategy. Many works suggest to sample closer nodes as *difficult* negative samples. For instance, MCNS [55] proposes that the negative sampling distribution should be positively correlated with the positive sampling distribution. KBGAN [4] samples most similar entities to replace the groundtruth positive entity in knowledge graph embedding and RecNS [56] tends to sample negative nodes from the intermediate distance region.

## 6 Conclusion and Limitations

In this paper, we study the sampling strategies of network embedding. To uncover the underlying relationships of existing competing sampling strategies, we conduct theoretical analysis on the sampling-based network embedding process. In the analysis, we identify two desirable properties for the similarity scores of node embedding, including the *discrimination* property and the *monotonicity* property. Furthermore, we prove that there always exists an error gap between the empirical and ideal optimal embedding similarity scores. Guided by such analysis, we propose a simple yet novel model (SENSEI), which creatively integrates the two competing sampling strategies to fulfill the the *discrimination* property and the *partial monotonicity* property. The effectiveness of SENSEI is verified by extensive experiments.

This paper studies sampling strategies of network embedding, which has no negative ethical impacts on society. The limitations of our paper lie in that the theoretical analysis is conducted on proximity scores and the proposed SENSEI model is designed for plain (non-attributed) networks rather than attributed networks. The key to generalize the proposed properties/SENSEI model to attributed networks is a new definition of the node pair similarity, which we leave for future exploration.

## 7 Acknowledgement

This work is supported by NSF (1947135, 2134079, 2316233, 1939725, and 2324770), DARPA (HR001121C0165), NIFA (2020-67021-32799), DHS (17STQAC00001-07-00), ARO (W911NF2110088). The content of the information in this document does not necessarily reflect the position or the policy of the Government, and no official endorsement should be inferred. The U.S. Government is authorized to reproduce and distribute reprints for Government purposes notwithstanding any copyright notation here on.

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

# A  Appendix

The appendix is structured as follows:

- Subsection A.1 gives the proof of Theorem 2;

- Subsection A.2 gives the detailed algorithm of SENSEI;

- Subsection A.3 introduces the statistics of datasets and the implementation details of SENSEI;

- Subsection A.4 compares the performance of SENSEI with 5 additional GCN baselines (Cluster-gcn, GraphSAGE, APPNP, GPRGNN and H2GCN) on Cora and Citeseer.

## A.1  Proofs

**Theorem 2.** *Mean Squared Error of $S$ between the Ideal Loss and the Empirical Loss*. *For the mean squared error between $S^*$ and $S_e$, we have*

$$\mathbb{E}[||(S_e - S^*)_u||^2] = \frac{1}{T(2 + \frac{kp_n(u|v)}{p(u|v)} + \frac{p(u|v)}{kp_n(u|v)})^2}\left(\frac{1}{p(u|v)} + \frac{1}{kp_n(u|v)} - 1 - \frac{1}{k}\right) \tag{13}$$

*where $\mathbb{E}$ is the expectation and $S^* = [\frac{p(u_1|v)}{p(u_1|v)+kp_n(u_1|v)}, \ldots, \frac{p(u_n|v)}{p(u_n|v)+kp_n(u_n|v)}]$.*

*Proof.* In this proof, since $S = [s(u_1, v), s(u_2, v), \ldots, s(u_n, v)]$ is the parameter to be optimized, we can write the empirical loss $J_e$ as a function of $S$ as $J_e(S)$. We prove this theorem with the help of Taylor expansion of $\nabla_S J_e(S_e)$ around $S^*$. Because $S_e$ is the solution to minimize $J_e$, $\nabla_S J_e(S_e) = \mathbf{0}$. So, it can be expressed as:

$$\nabla_S J_e(S_e) = \nabla_S J_e(S^*) + \nabla_S^2 J_e(S^*)(S_e - S^*) + O(||S_e - S^*||^2) = \mathbf{0} \tag{14}$$

Thus, up to terms of order $O(||S_e - S^*||^2)$, we obtain

$$S_e - S^* = -(\nabla_S^2 J_e(S^*))^{-1} \nabla_S J_e(S^*) \tag{15}$$

Then, we discuss the term $-(\nabla_S^2 J_e(S^*))^{-1}$ and the term $\nabla_S J_e(S^*)$ respectively.

For the term $-(\nabla_S^2 J_e(S^*))^{-1}$, let $\mathbf{e}_{(u_i)}$ be the one-hot vector, where only the $u_i$-th entry is 1.

$$J_e(S) = -\frac{1}{T}\sum_{i=1}^{T} \log(s(u_i, v)) - \frac{1}{T}\sum_{i=1}^{kT} \log(1 - s(w_i, v))$$

$$\nabla_S J_e(S) = -\frac{1}{T}\sum_{i=1}^{T} \frac{1}{s(u_i, v)}\mathbf{e}_{(u_i)} - \frac{1}{T}\sum_{i=1}^{kT} \frac{1}{s(w_i, v) - 1}\mathbf{e}_{(w_i)}$$

$$\nabla_S^2 J_e(S) = \frac{1}{T}\sum_{i=1}^{T} \frac{1}{s(u_i, v)^2}\mathbf{e}_{(u_i)}\mathbf{e}_{(u_i)}^{\top} + \frac{1}{T}\sum_{i=1}^{kT} \frac{1}{(s(w_i, v) - 1)^2}\mathbf{e}_{(w_i)}\mathbf{e}_{(w_i)}^{\top} \tag{16}$$

$$-\nabla_S^2 J_e(S) = -\frac{1}{T}\sum_{i=1}^{T} \frac{1}{s(u_i, v)^2}\mathbf{e}_{(u_i)}\mathbf{e}_{(u_i)}^{\top} - \frac{1}{T}\sum_{i=1}^{kT} \frac{1}{(s(w_i, v) - 1)^2}\mathbf{e}_{(w_i)}\mathbf{e}_{(w_i)}^{\top}$$

Since $s^*(u_i, v) = \frac{p(u_i|v)}{p(u_i|v)+kp_n(u_i|v)}$ and $s^*(w_i, v) = \frac{p(w_i|v)}{p(w_i|v)+kp_n(w_i|v)}$,

$$-\nabla_S^2 J_e(S^*) = -\frac{1}{T}\sum_{i=1}^{T} \frac{(p(u_i|v) + kp_n(u_i|v))^2}{p(u_i|v)^2}\mathbf{e}_{(u_i)}\mathbf{e}_{(u_i)}^{\top}$$

$$-\frac{1}{T}\sum_{i=1}^{kT} \frac{(p(w_i|v) + kp_n(w_i|v))^2}{(kp_n(w_i|v))^2}\mathbf{e}_{(w_i)}\mathbf{e}_{(w_i)}^{\top} \tag{17}$$

Therefore,

$$\lim_{T \to +\infty} -\nabla_S^2 J_e(S^*) = -\sum_{u_i} p(u_i|v) \frac{(p(u_i|v) + kp_n(u_i|v))^2}{p(u_i|v)^2} \mathbf{e}_{(u_i)} \mathbf{e}_{(u_i)}^\top$$
$$-\sum_{w_i} kp_n(w_i|v) \frac{(p(w_i|v) + kp_n(w_i|v))^2}{(kp_n(w_i|v))^2} \mathbf{e}_{(w_i)} \mathbf{e}_{(w_i)}^\top \qquad (18)$$
$$= -\sum_{u_i} \frac{(p(u_i|v) + kp_n(u_i|v))^3}{kp_n(u_i|v)p(u_i|v)} \mathbf{e}_{(u_i)} \mathbf{e}_{(u_i)}^\top$$

where the last line is because that we combine same terms when $u_i = w_i$.

Then, we consider the second term $\nabla_S J_e(S^*)$. We analyze the expectation and variance of $\nabla_S J_e(S^*)$ to get the final $\mathbb{E}[||(S_e - S^*)_u||^2]$.

For the expectation, as $\nabla_S J_e(S)$ is as follows:

$$\nabla_S J_e(S) = -\frac{1}{T} \sum_{i=1}^{T} \frac{1}{s(u_i, v)} \mathbf{e}_{(u_i)} - \frac{1}{T} \sum_{i=1}^{kT} \frac{1}{s(w_i, v) - 1} \mathbf{e}_{(w_i)} \qquad (19)$$

the expectation is:

$$\mathbb{E}[\nabla_S J_e(S^*)] = -\sum_{u_i} p(u_i|v) \frac{p(u_i|v) + kp_n(u_i|v)}{p(u_i|v)} \mathbf{e}_{(u_i)}$$
$$+ \sum_{w_i} kp_n(w_i|v) \frac{p(w_i|v) + kp_n(w_i|v)}{kp_n(w_i|v)} \mathbf{e}_{(w_i)} \qquad (20)$$
$$= \mathbf{0}$$

For the variance of $\nabla_S J_e(S^*)$, since $\mathbb{E}[\nabla_S J_e(S^*)] = \mathbf{0}$

$$\mathrm{Cov}(\nabla_S J_e(S^*)) = \mathbb{E}[\nabla_S J_e(S^*) \nabla_S J_e(S^*)^\top]$$
$$= \frac{T}{T^2} \mathbb{E}_{u \sim p(u|v)} \left( \frac{1}{s^*(u, v)^2} \right) \mathbf{e}_{(u)} \mathbf{e}_{(u)}^\top$$
$$+ \frac{kT}{T^2} \mathbb{E}_{w \sim p_n(w|v)} \left( \frac{1}{(1 - s^*(w, v))^2} \right) \mathbf{e}_{(w)} \mathbf{e}_{(w)}^\top$$
$$+ \sum_{u_i, u_j} \frac{T(T-1)}{T^2} p(u_i|v)p(u_j|v) \frac{1}{s^*(u_i, v)} \frac{1}{s^*(u_j, v)} \mathbf{e}_{(u_i)} \mathbf{e}_{(u_j)}^\top$$
$$- \sum_{u_i, w_j} \frac{kT^2}{T^2} p(u_i|v)p_n(w_j|v) \frac{1}{s^*(u_i, v)} \frac{1}{1 - s^*(w_j, v)} \mathbf{e}_{(u_i)} \mathbf{e}_{(w_j)}^\top \qquad (21)$$
$$- \sum_{u_j, w_i} \frac{kT^2}{T^2} p(u_j|v)p_n(w_i|v) \frac{1}{s^*(u_j, v)} \frac{1}{1 - s^*(w_i, v)} \mathbf{e}_{(w_i)} \mathbf{e}_{(u_j)}^\top$$
$$+ \sum_{w_i, w_j} \frac{kT(kT-1)}{T^2} p_n(w_i|v)p_n(w_j|v) \frac{1}{1 - s^*(w_i, v)} \frac{1}{1 - s^*(w_j, v)} \mathbf{e}_{(w_i)} \mathbf{e}_{(w_j)}^\top$$

Because $s^*(u_i, v) = \frac{p(u_i|v)}{p(u_i|v)+kp_n(u_i|v)}$, $s^*(u_j, v) = \frac{p(u_j|v)}{p(u_j|v)+kp_n(u_j|v)}$, $s^*(w_i, v) = \frac{p(w_i|v)}{p(w_i|v)+kp_n(w_i|v)}$ and $s^*(w_j, v) = \frac{p(w_j|v)}{p(w_j|v)+kp_n(w_j|v)}$, Eq. (21) becomes:

$$
\begin{aligned}
\mathrm{Cov}(\nabla_S J_e(S^*)) &= \frac{1}{T} \sum_u \frac{(p(u|v)+kp_n(u|v))^3}{kp_n(u|v)p(u|v)} \mathbf{e}_{(u)}\mathbf{e}_{(u)}^\top \\
&+ \sum_{u_i, u_j} (1 - \frac{1}{T})(p(u_i|v)+kp_n(u_i|v))(p(u_j|v)+kp_n(u_j|v))\mathbf{e}_{(u_i)}\mathbf{e}_{(u_j)}^\top \\
&- \sum_{u_i, w_j} (p(u_i|v)+kp_n(u_i|v))(p(w_j|v)+kp_n(w_j|v))\mathbf{e}_{(u_i)}\mathbf{e}_{(w_j)}^\top \\
&- \sum_{u_j, w_i} (p(u_j|v)+kp_n(u_j|v))(p(w_i|v)+kp_n(w_i|v))\mathbf{e}_{(w_i)}\mathbf{e}_{(u_j)}^\top \\
&+ \sum_{w_i, w_j} (1 - \frac{1}{kT})(p(w_i|v)+kp_n(w_i|v))(p(w_j|v)+kp_n(w_j|v))\mathbf{e}_{(w_i)}\mathbf{e}_{(w_j)}^\top \\
&= \frac{1}{T} \sum_u \frac{(p(u|v)+kp_n(u|v))^3}{kp_n(u|v)p(u|v)} \mathbf{e}_{(u)}\mathbf{e}_{(u)}^\top \\
&- (1 + \frac{1}{k})\frac{1}{T} \sum_{u_i, u_j} (p(u_i|v)+kp_n(u_i|v))(p(u_j|v)+kp_n(u_j|v))\mathbf{e}_{(u_i)}\mathbf{e}_{(u_j)}^\top
\end{aligned}
\tag{22}
$$

Based on Eq. (15), Eq. (18), Eq. (22) and $\mathbb{E}[||(S_e - S^*)_u||^2] = \mathrm{Cov}(S_e - S^*)(u, u)$, we have

$$
\begin{aligned}
&\mathbb{E}[||(S_e - S^*)_u||^2] = \\
&\frac{1}{T} \frac{kp_n(u|v)p(u|v)}{(p(u|v)+kp_n(u|v))^3} \left( \frac{(p(u|v)+kp_n(u|v))^3}{kp_n(u|v)p(u|v)} \right. \\
&\left. -(1 + \frac{1}{k})(p(u|v)+kp_n(u|v))^2 \right) \frac{kp_n(u|v)p(u|v)}{(p(u|v)+kp_n(u|v))^3}
\end{aligned}
\tag{23}
$$

which can be simplified as:

$$
\begin{aligned}
&\mathbb{E}[||(S_e - S^*)_u||^2] \\
&= \frac{1}{T} \frac{(kp_n(u|v)p(u|v))^2}{(p(u|v)+kp_n(u|v))^4} \left( \frac{kp_n(u|v)+p(u|v)}{kp_n(u|v)p(u|v)} - 1 - \frac{1}{k} \right) \\
&= \frac{1}{T(2 + \frac{kp_n(u|v)}{p(u|v)} + \frac{p(u|v)}{kp_n(u|v)})^2} \left( \frac{1}{p(u|v)} + \frac{1}{kp_n(u|v)} - 1 - \frac{1}{k} \right)
\end{aligned}
\tag{24}
$$

$\square$

## A.2 The Algorithm of SENSEI

In this section, we give the detailed algorithm of SENSEI in Algorithm 1.

## A.3 Dataset Statistics and Implementation Details.

In this section, we introduce the statstics of datasets and the implementation details of SENSEI. The statistics of datasets are present in Table 3.

Table 3: Dataset Statistics.

| Dataset | Layers | Nodes | Edges |
|---------|--------|-------|-------|
| C.ele | 1 | 297 | 2,148 |
| Cora | 1 | 2,708 | 5,429 |
| Citeseer | 1 | 3,327 | 4,732 |
| NS | 1 | 1,589 | 2,742 |

**Implementation Details.** For all the methods, we run 5 times to calculate the standard deviation and for all baselines, we use their default parameters. For SENSEI on 4 datasets: {C.ele, Cora, Citeseer,

**Algorithm 1** The algorithm of SENSEI

---

**Input:** (1) the input plain network $\mathbf{A}$; (2) training epochs in Step 1 ($E_1$) and in Step 2 ($E_2$), hyper-parameters $\tau$, $k$ and $\gamma$.
**Output:** Node embeddings $\mathbf{F}$.
  Initialize $\mathbf{F}$;
  **for** each $v$ in $\mathbf{A}$ **do**
    Run PPR to construct $\mathcal{P}(v)$ and $\mathcal{N}(v)$ with the threshold $\tau$;
  **end for**
  # Step 1: Fulfill Discrimination Property
  $e = 0$;
  **while** $e < E_1$ **do**
    **for** each $v$ in $\mathbf{A}$ **do**
      Randomly sample $k$ negative nodes from $\mathcal{N}(v)$;
      Compute $J(v)$ using Eq. (9);
    **end for**
    Add all $J(v)$s to obtain $J_P$ in Eq. (10);
    Back-propagate the loss and update $\mathbf{F}$;
    $e = e + 1$;
  **end while**
  # Step 2: Fulfill Partial Monotonicity Property
  $e = 0$;
  **while** $e < E_2$ **do**
    **for** each $v$ in $\mathbf{A}$ **do**
      Compute $L(v)$ using Eq. (11);
    **end for**
    Add all $L(v)$s to obtain $L_P$ in Eq. (12);
    Back-propagate the loss and update $\mathbf{F}$;
    $e = e + 1$;
  **end while**
  **return** $\mathbf{F}$.

---

NS}, we set the threshold $\tau$ as $\{0.008, 0.01, 0.005, 0.05\}$, the number of epochs in Step 1 as $\{40, 100, 20, 20\}$, the number of epochs in Step 2 as $\{40, 100, 50, 20\}$, the learning rate in Step 1 as $\{0.02, 0.1, 0.1, 0.2\}$, the learning rate in Step 2 as $\{0.01, 0.01, 0.005, 0.1\}$, the positive margin $\gamma$ as $\{0.05, 0.0001, 0.0001, 0.1\}$ and the negative sample number $k$ as 40 on all 4 datasets. All experiments are run on a Tesla-V100 GPU. [3]

### A.4 Additional Baselines of Plain Network Embedding

We have added another 5 GNNs as additional baselines: GraphSAGE [15], Cluster-gcn [5], APPNP [13], GPRGNN [6] and H2GCN [63]. the results are demonstrated in Table 4. We can see that for both link prediction and node recommendation, SENSEI generally beats these GNN methods.

Table 4: The AUC-ROC and AUC-PR of link prediction and Hit@10 of node recommendation on Cora and Citeseer(%).

| Models | Cora | | | Citeseer | | |
|---|---|---|---|---|---|---|
| | AUC-ROC | AUC-PR | Hit@10 | AUC-ROC | AUC-PR | Hit@10 |
| GraphSAGE [15] | 64.13 | 65.36 | 3.92 | 66.10 | 68.94 | 4.74 |
| Cluster-gcn [5] | 65.93 | 72.39 | 4.65 | 67.33 | 70.45 | 5.42 |
| APPNP [13] | 68.77 | 72.39 | 11.72 | 66.45 | 71.57 | 11.96 |
| GPRGNN [6] | 64.43 | 70.15 | 13.31 | 62.93 | 68.69 | 10.64 |
| H2GCN [63] | 61.90 | 61.89 | 1.96 | 60.95 | 61.94 | 3.53 |
| SENSEI | **81.27** | **83.59** | **20.51** | **76.28** | **82.02** | **19.03** |

---

[3]The simplified code of SENSEI is on: https://github.com/yucheny5/SENSEI.

