# OpenReview forum: "Reconciling Competing Sampling Strategies of Network Embedding"
_NeurIPS.cc/2023/Conference — NeurIPS 2023 poster_

### Official Review · Reviewer_FQQK · 2023-06-26

**Soundness:** 3 good
**Presentation:** 3 good
**Contribution:** 3 good
**Rating:** 7
**Confidence:** 3

**Summary:**

The paper first introduces two desirable properties for network embedding. The discrimination property expresses how well the model is able to distinguish node pairs that are similar from those which are dissimilar. The monotonicity property expresses how the similarity scores obtained from the embedding follow the same order as the probability of a pair in the original dataset. The authors first define these two properties, argue why they are important for network embedding methods, and illustrate them with an example.
Then, they consider a general framework for network embedding algorithm.

They present a standard expression of the likelihood of such models, discuss the elaboration of positive and negative samples sets, and derive elementary results of such models regarding the monotonicity and discrimination properties. They show both can be satisfied in an ideal case where infinite data is available (possibility results), and how both cannot be simultaneously satisfied for finite dataset (impossibility results). Doing so, they derive an optimal expression for the similarity between pairs of nodes, and how the positive sampling distribution should (negatively) correlate with the negative sampling distribution in order to satisfy monotonicity. Finally, they analyse the expected gap between empirical optimal solutions (finite dataset) and theoretical optimal solutions (infinite dataset).

They discuss the implication of these demonstrations. In particular, they detail some caveats of state-of-the-art sampling strategies in light of their findings.

Finally, they design a simple embedding procedure (SENSEI) that focuses on satisfying the discrimination and monotonicity properties to some extent. They show on 4 real-world datsets how their approach outperforms existing methods both in terms of link prediction (related to the discrimination property) and node recommendation (related to the monotonicity property).

They conclude the paper with a short overview of existing works on network embedding.

**Strengths:**

- The problem discussed is general; many previous works could be studied in the light of these findings
- Writing is clear and sections are well organized.
- Experiments are well carried out, with ablation tests, numerous baselines and four real-world datasets.
- The discussion of previous works limits in the light of some findings (Thm.2) is interesting and insightful.


**Weaknesses:**

- Illustrations could be improved. The first line of Fig.1 is not very helpful, gray bars are not explicitly defined, and the symbolic representation of p(u|v) is unclear. I believe simpler visualisations are doable. Also, figures should take the entire width of the page, in my opinion, instead of floating in the middle of the text.
- There are several mentions of "large" and "small" values of p and s. While it helps get a sense of what is discussed, I believe these should be better defined, as they are key words in the demonstrations. In line 233, the authors introduce the "intermediate" values with a rigorous definition; I believe discussing this above in the text might help to reach an explicit definition of "large" and "small".
- All demonstrations are in the appendix. I understand that available space is an issue for large mathematical formulas, but proofs for Th.1 and Prop.1, that are key in the text, are short enough to be included in the main text. In their absence, the main paper cannot be considered as self-sufficient.

**Questions:**

- l.232-233 - observing the deviation of p(u|v) entries could help choosing an optimal number for k?

- l.141 - Definition of discrimination property : because s \in [0;1] and p \in [0;1], the definition seems to boil down to s(w|v)->0 when p(w|v)->0. Also, the definition only considers extreme cases where two nodes are so dissimilar the ratios of p and s diverge; what about intermediate cases, such as the ones given in the paragraph under the definitions? What does "clearly distinguished in the embedding space" mean?

- l.144-166 - While I understand this is an illustrative example, I believe there should be at least a small discussion the chosen values of s(.,.). For instance, why would s(u1,v)<s(u2,v) given p(u1|v)>p(u2|v)? Does it happen often? Same for the very close values of s in case 2.

- Eq.7 - Why does k appear along with positive samples, when it is absent from Eq.4?

Additional comments:

l.107 - the link between discrimination/monotonicity and link pred/node rec should be stated higher in the text

Fig.1 - Maybe a matter of taste, but text should not wrap figures. The signification of the two-way gray arrows should be explicited in the legend. Besides, I am not sure that the figure helps the understanding of lines 144-166.

Eq.1 - I think the transition between Eq.1 and Eq.2 would be clearer if sigma(-F(w,:)^T F(v,:)) would be directly written as 1-sigma(F(w,:)^T F(v,:)). Also, there should be a reference to Def.1, because some s(.,.) could take values that are not in [0,1]. Dissimilar vector can yield a negative cosine similarity, for instance.

Appendix A.2, Th.1 - Because there is no space constraint, it would be clearer to explicit the definition of the expectation E before differentiating.

Appendix A.2, Th.2, l.600 - Typo "is is".

l.206 - Specify the theorem holds for a 1st order Taylor expansion.

l.211-220 - Nice discussion.

l.241-256 - Fig.1 should be placed and explained here. Plotting fictive values of p(u|v) on top of the s(.,.) lines (instead of symbols) would ease the understanding.

Tables - Results with overlapping std should be emphasized in a similar way (bold or underline)

**Limitations:**

In the words of the authors, with which I agree:
"The limitations of our paper lie in that the theoretical analysis is conducted on proximity scores and the proposed SENSEI model is designed for plain (non-attributed) networks rather than attributed networks. The key to generalize the proposed properties/SENSEI model to attributed networks is a new definition of the “proximity” (i.e., node pair similarity). Currently, the proximity refers to the topological aspect of the node pair similarity."
These limitations lay the ground for broader future theoretical works on the topic.

---

> ### Author Rebuttal · Authors · 2023-08-09
>
> **Thanks: First, we want to thank you again for your *very detailed comments and many valuable suggestions*, which are very helpful for us to further improve the readability and quality of our paper in the revised version.**
>
> **Response to Q1**:
>
> This is a good observation that the deviation of $p(u|v)$ and $k$ will together determine whether the node $u$ belongs to the group of nodes with largest/smallest/intermediate $p(u|v)$s and whether the similarity function $s(u,v)$ can be well learned. However, we are afraid that we can not say that "the deviation of $p(u|v)$ entries could help choosing an *optimal* number for $k$". This is because the number for $k$ is usually manually chosen for the *whole* graph and our analysis is based on one specific center node $v$. A good $k$ for one specific center node $v$ may not be a good $k$ for another center node $v^{\prime}$ in this graph. Thus, we can not claim that `"the deviation of $p(u|v)$ entries could help choosing an *optimal* number for $k$''. We think that $k$ should be tuned as a hyper-parameter for different datasets.
>
> **Response to Q2**:
>
> First, it is true that the definition will force $s(w|v)$ to approach 0 when $p(w|v)\rightarrow0$.
>
> Second, for these intermediate cases and the definition of discrimination property, we had considered adopting a threshold based definition: with two given proximity thresholds $0<p\_{1}<p\_{2}<1$ and two similarity score thresholds $0<s\_{1} < s\_{2}< 1$, if $p(u|v) < p\_{1}$, it is regarded as "small" and $s(u, v)$ should be smaller than $s\_{1}$. If $p\_{1} \le p(u|v) \le p\_{2}$, it is regarded as "intermediate" and the range for $s(u, v)$ should fall into $[s\_{1}, s\_{2}]$. If $p(u|v)>p\_{2}$, it is regarded as "large" and $s(u, v)$ should be larger than $s\_{2}$. However, we eventually decided to use the current definition presented in the paper for the following reasons:
>
> (1) the threshold is a *hard boundary* to discriminate between large/intermediate/small but  the range of $p(u|v)$ is continuous. For example, if the threshold $p\_{2}=0.3$, it is unreasonable to say that $p(u\_{1}|v)=0.300001$ is large while $p(u\_{2}|v)=0.299999$ is intermediate.
>
> (2) the thresholds should be decided case by case for different center node $v$s in the graph. It is likely that one threshold value may be a good choice as $p\_{1}$ for center node $v$, but for another center node $v^{\prime}$, the threshold is more suitable to function as $p\_{2}$.
>
> (3) the *limitation* style definition in l.141 is consistent with our following discussion and analysis, which will make it easier for the readers to understand the key ideas/findings of our paper.
>
> Third, for "clearly distinguished in the embedding space", this means that for the link prediction task, we want the embedding similarity scores of node pairs with/without links to be much larger (e.g., $s(u,v)> 0.7$)/smaller (e.g., $s(u,v)< 0.3$) than the classification/decision boundary (e.g., $0.5$).
>
> Thank you again for raising the discussion about the definition of the discrimination property, we will add the above discussion into the main contents in the revised version of our paper.
>
> **Response to Q3**:
>
> Thanks for the suggestion.
>
> For the first question, it often happens when we adopt the empirical objective function in Eq.4. If $p(u\_{1}|v)>p(u\_{2}|v)$ but the values are close, due to the limited number of positive samples $T$, it is possible that $u\_{2}$ is sampled more times (e.g., once or twice) than $u\_{1}$ in the sampling process, which will cause $s(u\_{1},v)<s(u\_{2},v)$.
>
> For the second question related to the very close values of s in case 2, this can appear in hard negative sampling strategies. Some nodes with large/intermediate $p(u|v)$s will be sampled as difficult negative samples due to the hard negative sampling strategies. This will make these nodes have small similarity scores w.r.t. the center node, which can not be distinguished with nodes with small $p(u|v)$s. We will add the above discussion into the revised paper.
>
> **Response to Q4**:
>
> This is an implementation trick: it is equivalent to repeatedly sampling one specific node multiple times. In this way, we can make the best of the positive samples to make the training process converge more quickly.
>
>  **Response to Weakness 1**:
>
> Thanks for the great suggestion. We will separate the first line (case 1 and case 2) with the second and third line of Figure 1. We will also follow the reviewer's suggestion to add some fictive values of $p(u|v)$ on top of the s(.,.) lines " in Figure 1.  **Please see the attached figure file uploaded in the general rebuttal.**
>
> **Response to Weakness 2**:
>
> Please refer to our response to **Q2** about the discussion of the definitions. In addition, we will incorporate some notations and discussions similar to line 233 (e.g. $p(u|v) >> kp\_{n}(u|v)$ and $p(u|v) << kp\_{n}(u|v)$) in the revised version of our paper.
>
> **Response to Weakness 3**:
>
> We will move proofs for Th.1 and Prop.1 to the main text in the revised version of our paper.
>
> **Response to Additional Comments**:
>
> Thank you again for your very detailed and valuable comments on these typos/minor points. We will follow the reviewer's suggestion to fix the typos/minor points and polish the paper in the revised version accordingly.

---

> > ### Comment · Reviewer_FQQK · 2023-08-10
> > **OK**
> >
> >  I acknowledge that I have read the reviews and responses. My rating stays the same.

---

> > > ### Author Response · Authors · 2023-08-11
> > > **Thanks for the acknowledgement of reading our responses**
> > >
> > > Thanks again for your valuable comments and the acknowledgement of reading our responses.

---

### Official Review · Reviewer_1EU9 · 2023-07-03

**Soundness:** 3 good
**Presentation:** 4 excellent
**Contribution:** 3 good
**Rating:** 7
**Confidence:** 4

**Summary:**

This work targets on network embedding under different strategies. The authors first theoretically demonstrate the dilemma in existing network embedding algorithms, especially the sampling-based framework, that the sampling-based embedding performance can have an error gap due to the competing sampling strategies.  Then, they propose two properties, i.e., discrimination and monotonicity, that should be obeyed, and eventually, under this perspective, they propose a novel network embedding approach called SENSEI to perform the (plain) network embedding. The SENSEI includes two steps, where the first step samples nodes with intermediate distribution following the first category of sampling strategies to satisfy the discrimination property, while the second step follows the second category of sampling strategies to partially satisfy the monotonicity property. Experiments on several datasets show the effectiveness of the proposed approach.

**Strengths:**

+ This work has a clear motivation to devise a new reconciling strategy to satisfy the discrimination and monotonicity properties.
+ The theoretical analysis is sound and details the significance of the two steps. Corresponding discussions clearly tell the advantages of the proposed embedding approach.
+ The performance of SENSEI is not incremental in both link prediction and node recommendation tasks. Though the work mainly performs on plain networks, the authors still give the results on multi-layer networks.
+ The proposed approach seems not to be difficult to reproduce.


**Weaknesses:**

- This work can only be effective on sampling-based embedding algorithms, though I think it is valuable for this field.
- According to the results of Table 2, the promotion of step 2 is incremental. However, will step 2 harm the performance of the link prediction task? (See my second question in `Questions’) If so, the necessity of reconciling the competing sampling strategies is not very strong.


**Questions:**

+ Why is this approach called `SENSEI’?
+ According to intuition, the SENSEI-P (w/o Step 2) will perform better than SENSEI-P on the link prediction task. Is it right?


**Limitations:**

Yes. The authors claim that the attribute could have helped promote the approach.

---

> ### Author Rebuttal · Authors · 2023-08-08
>
> **Q1: Why is this approach called 'SENSEI’?**
>
> **Response**: Thanks for asking us the question about the name of our model. Here we adopt 'SENSEI' for its similarity to the word 'sense'. The word 'sense' has two levels of meanings: (1) the first level meaning is 'notice'. Our work notices the competing sampling strategies in existing methods; (2) the second level meaning is 'understand/unveil'. Our work understands and unveils the underlying reason behind such competing designs of sampling strategies with the help of sound theoretical analysis.
>
> **Q2: According to intuition, the SENSEI-P (w/o Step 2) will perform better than SENSEI-P on the link prediction task. Is it right?**
>
> **Response**: We are afraid  this is a factual misunderstanding. Step 2 will not hurt SENSEI-P's performance on link prediction. STEP 2 focuses on the partial monotonicity property within the positively sampled node set from STEP 1 to improve the performance in node recommendation task. In implementation, STEP 2 just fine-tunes the rank of node pairs with large similarity scores under a *smaller* learning rate. We use the toy example from Figure 1 in our paper to illustrate the optimization process: Assume that (1) $p(u\_{1}|v)=0.45$, $p(u\_{2}|v)=0.4$ and $p(w|v)=0.001$; (2) $u\_1$ and $u\_2$ have link connected with $v$, and $w$ does not have link connected to $v$. After Step 1, the similarity scores are likely be optimized to $s(u\_1,v)=0.7$, $s(u\_2,v)=0.8$ and $s(w,v)=0.01$. In Step 2, $s(u\_1,v)$ is fine-tuned to be 0.76, $s(u\_2,v)$ is fine-tuned to be 0.73 and $s(u\_1,v) > s(u\_2, v)$, and $s(w, v)$ is still 0.01. After Step 2, only the ranks of $u\_1$ and $u\_2$ are affected and the decision threshold (e.g.,$s(\cdot,\cdot) = 0.5$) for link prediction is not affected.
>
> **Other points in weaknesses/limitations/comments**:
>
>  **P1: This work can only be effective on sampling-based embedding algorithms, though I think it is valuable for this field.**
>
> **Response**: First, as we mentioned in the introduction section and the related works section: "many existing network embedding methods follow a sampling training procedure." Please refer to these two sections for more details. Second, we would like to make the following clarification on the generality of our work. The essence of link prediction task is a binary classification task and the essence of node recommendation task is a ranking task. Thus, our theoretical findings can be transferred to proximity/similarity based binary classification task and ranking task in other domains (e.g., recommender system/NLP) rather than the graph embedding domain. Thus, we believe that our work can be generalized to a broad range of similar tasks in other domains.

---

> > ### Comment · Reviewer_1EU9 · 2023-08-13
> > **Response to authors**
> >
> > Thanks for the authors' response. I have read the responses and reviews and I keep the current rating.

---

> > > ### Author Response · Authors · 2023-08-13
> > > **Thanks**
> > >
> > > Thanks for reading our response and supporting our paper!

---

### Official Review · Reviewer_fuvb · 2023-07-11

**Soundness:** 3 good
**Presentation:** 3 good
**Contribution:** 4 excellent
**Rating:** 8
**Confidence:** 4

**Summary:**

Graph embedding methods typically involve negative sampling, where a set of nodes are sampled as positive samples (usually nodes of close proximity, such as edges) and a set of nodes are sampled as negative samples. While early work tended to randomly sample negatives from non-edges, more recent work has adopted different approaches for negative sampling. The authors identify two main classes of methods for negative sampling in the literature. Their main contribution is identifying two properties that they claim any good set of node embeddings should satisfy: discrimination and monotonicity. They then demonstrate that these two properties cannot be simultaneously satisfied with a finite number of samples. They propose the SENSEI method that satisfies discrimination and a *partial* monotonicity property and demonstrate strong empirical performance.

*After rebuttal:* The authors have clarified my concerns, and I continue to support the paper. I also did not see anything from the other reviews that would negatively affect my opinion.

**Strengths:**

- Provides an explanation unifying different negative sampling strategies for graph embeddings based on different desirable properties. This is a novel way of thinking about the graph embedding problem.
- Interesting theoretical results with practical applicability: a possibility theorem to satisfy the two desirable properties in with infinite samples along with an *impossibility* theorem to satisfy the two desirable properties in practice with finite samples.
- New SENSEI algorithm that satisfies a weaker version of the two desirable properties and shows strong empirical performance.

**Weaknesses:**

- Generality of results is questionable. The proposed properties of discrimination and monotonicity are very much tied to two assumed tasks: link prediction and node recommendation. If one is only interested in one of these tasks (or perhaps a different task altogether), then the trade off between these two properties may not apply.
- Discussion of limitations should be moved to the main paper.


Typos and minor issues:
- Figure 1 appears very early on in the paper, before the detailed discussion on the first and second category of methods on page 6. I did not understand it until reading page 6. I suggest the authors place a shorter summary of the first and second categories in the figure caption or earlier in the paper around the first time Figure 1 is used.

**Questions:**

1. The analysis in Section 2 is quite general with respect to the similarity function $s(u,v)$. Sigmoid and cosine functions are mentioned as examples. What requirements must the similarity function satisfy in order for your analysis to be valid? Would, for example, a Euclidean distance-based similarity such as $\sigma(||\mathbf{F}(u,:) - \mathbf{F}(v,:)||_2)$ meet the requirements?

2. It seems that the two properties of discrimination and monotonicity are very much tied to the two tasks that you assume are important: link prediction and node recommendation. However, if one is only interested in link prediction, is there a reason they should care about monotonicity rather than just discrimination?

**Limitations:**

Limitations are discussed, but not in much detail and only in the supplement. Such a discussion should be moved to the main paper, with a longer discussion in the supplement if necessary.

---

> ### Author Rebuttal · Authors · 2023-08-08
>
> **Q1: The analysis in Section 2 is quite general with respect to the similarity function $s(u,v)$. Sigmoid and cosine functions are mentioned as examples. What requirements must the similarity function satisfy in order for your analysis to be valid? Would, for example, a Euclidean distance-based similarity such as $\sigma(||\mathbf{F}(u,:)-\mathbf{F}(v,:)||_{2})$ meet the requirements?**
>
> **Response**: (1) Generally, the requirements for the similarity function are $s(u,v)$ is symmetric and $s(u,v)\in(0,1)$. Here we do not take the corner case that $s(u,v)=0$ or $s(u,v)=1$ because it will leads to $\log 0$ in the loss function. However, the theoretical conclusions still apply to $s(u,v)=0$ or $s(u,v)=1$ in implementation. (2) if $\sigma$ in $\sigma(||\mathbf{F}(u,:)-\mathbf{F}(v,:)||\_{2})$ is the sigmoid function, it still meets the requirement. In addition, we can adopt $e^{-||\mathbf{F}(u,:)-\mathbf{F}(v,:)||_{2}}\in(0, 1]$ as the similarity function, which is also Euclidean based and meets the requirement.
>
> **Q2: It seems that the two properties of discrimination and monotonicity are very much tied to the two tasks that you assume are important: link prediction and node recommendation. However, if one is only interested in link prediction, is there a reason they should care about monotonicity rather than just discrimination?**
>
> **Response**: First, the monotonicity property is closely related with the ranking of target similarity scores. If one is only interested in link prediction, this means that nodes are classified into two groups: (1) nodes with a link to the center node and (2) nodes without a link to the center node. In each group the target similarity scores of each node to the center node are regarded to be equal (1 or 0 respectively) since ''one is only interested in link prediction'' rather than node recommendation. In this case, the monotonicity does not need to be considered. Second, we would like to point out that the essence of link prediction task is a binary classification task and the essence of node recommendation task is a ranking task. Thus, our theoretical results can be transferred to proximity/similarity based binary classification task and ranking task in other domains (e.g., recommender system/NLP). Usually, both the discrimination property and the monotonicity property are important in real-world applications. For example, in an e-commerce system, predicting the candidate set of products that customers may have interests in is a binary classification task (discrimination property) and the rank of items within the candidate set also matters (monotonicity property). Our theoretical analysis is still useful in such real-world applications.
>
> **Other points in weaknesses/limitations/comments**:
>
> **P1 in Weakness 1: Generality of results is questionable. The proposed properties of discrimination and monotonicity are very much tied to two assumed tasks: link prediction and node recommendation. If one is only interested in one of these tasks (or perhaps a different task altogether), then the trade off between these two properties may not apply.**
>
> **Response**: As mentioned in our response to **Q2**, the essence of link prediction task is a binary classification task and the essence of node recommendation task is a ranking task. Thus, our theoretical findings can be transferred to proximity/similarity based binary classification task and ranking task in other domains (e.g., recommender system/NLP) rather than the graph embedding domain. Thus, we believe that these two properties are relevant to a broad range of similar tasks in other domains.
>
> **P2 in Weakness 2: Discussion of limitations should be moved to the main paper.**
>
> **Response**: Thanks for the suggestion. We will move the discussion of limitations to the main paper in the revised version of our paper.
>
> **P3 in Weakness 3: Figure 1 appears very early on in the paper, before the detailed discussion on the first and second category of methods on page 6. I did not understand it until reading page 6. I suggest the authors place a shorter summary of the first and second categories in the figure caption or earlier in the paper around the first time Figure 1 is used.**
>
> **Response**: Thanks for the suggestion. We will add the following text into the caption of Figure 1: ''For the first category of methods, they hurt the monotonicity between the star nodes and the round nodes. For the second category of methods, they keep the *monotonicity* between the star nodes and the round nodes but one rectangle node with small $p(u|v)$ has intermediate $s(u_8,v)$, which hurts the discrimination property.'' In addition, we will adjust the position of Figure 1 in the revised version to make it easier for readers to follow and understand the contents of our paper.

---

> > ### Comment · Reviewer_fuvb · 2023-08-21
> >
> > Thanks for the clarifications. I continue to support the paper and hope to see it accepted!

---

### Author Rebuttal · Authors · 2023-08-09

We sincerely thank all reviewers for their valuable time and insightful feedbacks which are very helpful for further improving the quality of our paper. We are grateful that the reviewers appreciate the novelty of the idea (`fuvb`), the importance of our work for "unifying existing different sampling strategies for graph embedding" (`fuvb`) and the potential of our paper for "broader future theoretical works on this topic" (`FQQK`). We are also encouraged that (1) our theoretical analysis is regarded as interesting (`fuvb`), insightful (`FQQK`) and sound (`1EU9`); (2) our experiments are well-carried out (`FQQK`) and the empirical performance is strong (`fuvb`, `1EU9`).  We have provided our point-to-point response to the questions of each reviewer below and we sincerely invite the reviewers for further discussion.

In addition, we split the first line with the second and third line in Figure 1 and we have followed the suggestion of Reviewer `FQQK` to modify Figure 1, which is presented in the attached pdf file.

---

### Decision · Program_Chairs · 2023-09-21

**Decision:**

Accept (poster)

**Comment:**

All reviewers agree that this paper is a good contribution. I recommend this paper to be accepted as a poster.